# Optimizing Glass Fiber Molding Process Design by Reverse Warping

**DOI:** 10.3390/ma13051151

**Published:** 2020-03-05

**Authors:** Han-Jui Chang, Zhi-Ming Su

**Affiliations:** Key Laboratory of Intelligent Manufacturing Technology, Shantou University, Ministry of Education, Shantou 515063, China; 19zmsu@stu.edu.cn

**Keywords:** glass fiber, reverse warpage, optimize molding, holding pressure

## Abstract

The purpose of this study is to clarify the influence of changes in glass fiber properties on warpage prediction, and to demonstrate the importance of accurate material property data in the numerical simulation of injection molding. In addition, this study proposes an optimization method based on the reverse warping agent model, in which the thermal conductivity of the plastic material is reduced, and the solidified layer on the surface of the mold is reduced and transferred from the molding material to the mold wall. This means that by the end of the cooling phase, the shrinkage of the molten zone within the component will continue, resulting in warpage. Based on the optimal process parameters, the sensitivity of the warpage prediction to the relationship between the two most important material properties, the glass fiber and holding pressure time, was analyzed. The material property model constants used for numerical simulations can sometimes vary significantly due to inherent experimental measurement errors, the resolution of the test device, or the manner in which the curve fit is performed to determine the model constants. This model has been developed to intelligently determine the preferred processing parameters and to gradually correct the details of the molding conditions. Thus, the cavity is separated in the critical warpage region, and a new cavity geometry with a reverse warped profile is placed into the mold base slot. The results show that the hypothetical and reasonable variation of the glass fiber model constant and the holding pressure time relationship may significantly affect the magnitude of the warpage prediction of glass fiber products. The greatest differences were found as a result of the warping orientation of the glass fiber material.

## 1. Introduction

The aesthetics and comfort of automotive parts have received more and more attention along with the development of automobiles. A large number of automobiles’ internal and external plastic parts and components are manufactured by injection molding, but these injection-molded parts often have warping deformation defects. Qualitative analysis can be used to optimize the process parameters to reduce the degree of warpage deformation in injection-molded parts. However, it is still difficult to accurately determine the actual amount of warpage deformation of the injection-molded parts without experimentation.

This paper studies the interior automotive parts made from polypropylene (PP) thermoplastic elastomers. The accurate prediction system of warpage deformation in injection-molded material parts was developed using current simulation software. The warpage deformation of PP injection-molded parts simulation was completed in two steps to avoid fluid-structure coupling and thermo-mechanical coupling calculations, thus simplifying the finite element model. The mold flow analysis is used to obtain the temperature distribution of the injection-molded parts at different times to establish the warping deformation finite element model; the temperature distribution obtained in the first step is used as the boundary condition. Improved PP material properties (VP switching performance and constitutive relation) were embedded in the warped deformation finite element model, and the key material properties of the warpage deformation of the injection-molded parts were successfully calculated. For glass fiber reinforced composite materials, the orientation distribution of long fibers in injection-molded parts was measured.

The demand for lightweight materials technology has grown in recent years, especially in automotive applications where fiber-reinforced thermoplastics have replaced specific metal parts because they are significantly lighter. In addition, the mechanical properties of fiber-reinforced thermoplastics can be greatly improved when compared to other injection-molded products due to their microstructure (such as orientation, length, and concentration). Long fiber-reinforced thermoplastics have become one of the most popular materials for lightweight applications, and the fiber structure plays a crucial role in the mechanical properties of reinforced composite materials. However, complete control of fiber properties (such as the increased strength of the material) is difficult because the microstructure of the fibers in the matrix is very complex and the structure and operation of the injection-molded screw can have a significant effect on the residual fiber length.

## 2. Literature Review

In 2019, Antun Maldini, Niels Samwel, Stjepan Picek, and Lejla Batina [1] proposed some solutions and included a novel and original algorithm for optimizing the parameter search for fault injection, with only some very general hypotheses for internal work that was superior to all known search methods. In 2017, Kun Li, Shilin Yan, Wenfeng Pan and Gang Zhao [2] mentioned that warpage is an important indicator for measuring the quality of parts in short fiber reinforced composite injection molding. In addition to the normal process parameters, fiber parameters also have a significant impact on part warpage. In this study, the objective function was the smallest issue of warping. Design parameters included the fiber aspect ratio, fiber content, injection time, melt temperature, mold temperature, and holding pressure. In 2017, Sigit Yoewono, Martowibowo and Agung Kaswadi [3] mentioned that the start of the optimization process involved setting machine parameters, namely the melting temperature, the injection pressure, the holding pressure, and the holding time. The genetic algorithm approach and Moldflow were used to optimize injection process parameters at minimum cycle times. It was found that the optimum injection molding process can be obtained by setting the parameters to the following values: TM = 180 °C; Pinj = 20 MPa; Phold = 16 MPa, Thold = 8 s, and the cycle time was 14.11 s. Further research using a conformal cooling system produced an average cycle time of 14.19 s. The conformal cooling system studied produced a volumetric shrinkage of 5.61% and found a wall shear stress of 0.17 MPa. In 2017, Kuo-Ming Tsai, and Hao-Jhih Luo [4] stated that the Taguchi parameters can be used to design screening experiments of injection molding parameters, where the important factors that affect lens shape accuracy are mold temperature, cooling time, packaging pressure, and packaging time. These important factors are used in full factorial experiments, from which the experimental data was then used as the training and inspection data set for the artificial neural network (ANN) prediction model. Finally, the ANN prediction model was combined with a genetic algorithm (GA) to construct an inverse model for injection molding. In addition, the global search algorithm used GA as the optimal solution to further optimize Taguchi’s optimal process parameters. Verification experiments demonstrated that the shape accuracy of the lens had improved. In 2016, Sheng-Liang Chen, Hoai-Nam Dinh, and Van-Thanh Nguyen [5] proposed the identification of suitable injection times, screw positions, and cavity pressures to transition from injection speed control to filler pressure control (often referred to as the filler transition point) that is critical for high quality parts. This method, that is based on the switching of the holding pressure in the mold cavity, obtains good results. The holding pressure process and the filling process are the most important stages when regarding part quality, with the corresponding key process variables being the injection speed and the filling pressure. In 2015, Jian Zhao, Gengdong Cheng, Shilun Ruan, and Zheng Li [6] raised the two-stage optimization system that is proposed in this study. In the first stage, an improved effective global optimization (IEGO) algorithm was used to approximate the non-linear relationship between the parameters and part quality measurements. In the second stage, the nondominated sorting genetic algorithm II (NSGA-II) was used to find better design solutions and better convergence near the true Pareto optimality. The cover of a liquid crystal display section was optimized to display this method. In 2016, Dyi-Cheng Chen and Chen-Kun Huang [7] mentioned combining the research analysis method and the Taguchi method to study injection molding warping by aggregating and summarizing the hierarchical data from the collected documents. This data then underwent statistical analysis to determine the optimal combination of the four factors of injection molding, injection pressure, holding pressure, holding time, and mold temperature with three levels of Taguchi design data to obtain the optimal combination for minimum warpage. In 2014, Wei Guo, Lin Hua, and Huajie Mao [8] referred to the impact rate of each factor and studied the interaction of each variable with the sink mark. The prediction model of the sink mark was effectively coupled with the GA to optimize variables to minimize the sink depth. The results of comparative analysis showed that the proposed method could be effectively used to minimize the sink mark dent depth and optimize the parameter design. In 2013, Juan Pablo Caballero-Villalobos, Gonzalo Enrique Mejía-Delgadillo, and Rafael Guillermo García-Cáceres [9] mentioned that as part of the modeling phase, a framework that proposes a mixed integer linear programming formula. There are two ways to make a chess model at the Columbia Company: introducing a suggested normalization operator to improve the results by reducing the search space of the GA, and by explaining the special constraints described above through PN modeling and using the GA to encoding the chromosomes. In 2012, Jin-ping Zhou and Hu Fu [10] proposed a two-stage genetic algorithm to manage the complexity of injection molding scheduling: the first stage is to allocate work to machines, and the second stage is to sort the work for each machine. A simulation model was proposed to solve the injection shop scheduling problem. To determine the optimal startup time of a single machine, a rule-based heuristic algorithm was also proposed. The application proves the reliability and effectiveness of algorithms and simulation models. In 2014, Dyi-Cheng Chen, Tse-Hsi Chen, Geng-Fu Lin, Yi-Kai Wang, and Yu-Chen Chang [11] conducted an experimental study on the effect of process parameters on the surface quality of injection-molded films using thermoplastic part sheets. The research focused on the shape, number, and location of mold gates, injection pressure, and injection rate. It was determined that the gravity-assisted entry point had improved cavity filling at the same forming time and injection pressure. In 2019, Kuna Li, Shilina Yan, Yuchenga Zhong, Wenfeng Pan, and Gang Zhao [12] stated that warpage, volume shrinkage, and residual stress are three important indicators of the overall quality of short fiber reinforced composite parts in the injection process. In addition to process parameters, fiber parameters also have a significant impact on part quality; according to the simulation results, three response surface models were created to plot the complex nonlinear relationship between the design parameters and quality goals. Based on the response surface model, the relationship between fiber parameters and quality indicators was further discussed. In 2010, Jin Wang and Xiaoshi Jin [13] noted that the Reduced Strain Closure model captures slow fiber orientation dynamics, a phenomenon that is observed in experiments, but the widely used the Folgar–Tucker model over-predicts. The model should take fiber-fiber interaction into account by the using anisotropic diffusion to correct the problem of long fiber materials, as the model cannot match all aspects of the measured fiber orientation data. The concept of the interaction effect is explained in physics as when two or more independent variables exist in an experimental study, and the effect of one of the independent variables exhibits an alternating phenomenon on the other independent variable. The true effect of some factors is changed by another factor. The variables separable model (VSM) in Taguchi’s method is used to evaluate this interaction effect. It can be used to quantify the interaction between the values of the warping orientation of glass fibers. This method provides reference data and provides a simple way to determine the amount of impact on the molded object depending on whether or not glass fiber was added and the holding pressure time condition [14,15,16].

## 3. Research Purpose

Products with high-precision dimensions can be assembled with certain tolerances, so the relative size of the product’s warpage is important. The subject of this experiment is the seat belt buckle part from the interior of an automobile. PP is used as the base material and different glass fiber ratios (0%, 20%, 30%) are added to the material to determine how the glass fiber ratios affect the amount of warpage deformation. There are two main factors involved in achieving the goal of warpage improvement: the proportion of glass fiber and the holding time. However, due to the interactions between these two factors in terms of varying proportions and time, measuring the data obtained from a single factor is not meaningful, as the effect of each factor must be studied at different proportions of the other factor. Therefore, the main purpose of this study is to clarify the interactions between the glass fiber ratio, the holding time, and the warpage deformation, as well as the presence of different warping orientations.

The interactions between the amount of warpage deformation and the proportion of glass fiber in the forming process are rarely discussed. When the interaction exists, it is meaningless to simply study the effect of one factor, as it is necessary to investigate the method of action of that factor at different levels. If there is only one influencing factor in all experimentally designed cells, the interaction between independent variables cannot be measured, such as when the interaction in the experimental design method indicates that an effect is stronger or weaker when two or more factor levels are acting simultaneously compared to the effect of a single factor level. In other words, the warping orientation also changes due to different proportions of glass fiber addition. Therefore, in this paper, when the two sets of data interact under the same forming conditions, and there are factor effects during different geometric parameter conditions, the actual and theoretical values obtained due to the interaction effects will also be different.

## 4. Fiber Orientation Control Model

Currently, many plastic products have entered the development production stage. Using glass fiber in plastic reduces the weight and improves durability and impact resistance, which are important inspection items of current product specifications. The main parameters during the addition of glass fiber filling flow that lead to deformation and structural strength of the product are glass fiber orientation, glass fiber length (glass fiber fracture prediction), and glass fiber concentration. There has been recent progress in the research on the influence of two parameters of glass fiber orientation and glass fiber length prediction. Through simulation analysis, the effects of these two factors on warpage deformation and product rigidity can be predicted; simulation analysis technology is a fast and powerful lightweight evaluation method to help users obtain more complete and accurate analysis results during the simulation phase, which help improves production efficiency.

In generalized vector fields, we do not adhere to every point in space. Each fiber is a medium, and we discuss different types of behavior. The fiber is regarded as the unit vector in mathematics, and it is assumed that the fiber flows perfectly along the *x*-axis, but the flow system includes the flow orientation, the crossflow direction, and the thickness orientation as [17]
(1)$$p=(\cos\alpha ,\cos\beta ,\cos\gamma ),\;0\leq \alpha ,\beta ,\gamma \leq \frac{\pi }{2}$$
(2)$$p=(1,0,0),\;(\alpha ,\beta ,\gamma )=(0,\frac{\pi }{2},\frac{\pi }{2})$$
(3)$$\cos^{2}\alpha +\cos^{2}\beta +\cos^{2}\gamma =1$$

Definition of fiber orientation can be seen in Figure 1.
(4)$$Α=pp=\left[\begin{array}{ccc} \cos^{2}\alpha & \cos\alpha \cos\beta & \cos\alpha \cos\gamma \\ \cos\alpha \cos\beta & \cos^{2}\beta & \cos\beta \cos\gamma \\ \cos\alpha \cos\gamma & \cos\beta \cos\gamma & \cos^{2}\gamma \end{array}\right]=\left[\begin{array}{ccc} A_{xx} & A_{xy} & A_{xz}\\ A_{xy} & A_{yy} & A_{yz}\\ A_{xz} & A_{yz} & A_{zz} \end{array}\right]$$
(5)$$Α=\left[\begin{array}{ccc} A_{11} & A_{12} & A_{13}\\ A_{12} & A_{22} & A_{23}\\ A_{13} & A_{23} & A_{33} \end{array}\right]$$
(6)$$A_{xx}+A_{yy}+A_{zz}=1$$
(7)$$\left(A_{xx},A_{yy},A_{zz})=(1,0,0\right)$$
(8)$$A_{11}+A_{22}+A_{33}=1$$
(9)$$0\leq A_{33}\leq A_{22}\leq A_{11}\leq 1$$

(Probability of a fiber direction along three different axes).

In 2010, Jin Wang* and Xiaoshi Jin [13] used the reduced strain closure (RSC) model to capture the slow fiber orientation dynamics that were observed in experiments, which the widely used Folgar–Tucker model over-predicts. The ARD model resolves the fiber-to-fiber interaction through anisotropic diffusion, which corrects the problem with the Folgar–Tucker model that long-fiber materials cannot match all aspects of the measured fiber orientation data.

Wang and Tucker proposed a new RSC model, which slows down the eigenvalue rate and fixes the eigenvectors of the orientation feature. The two parameters: the fiber interaction parameter CI and the deceleration parameter kappa, need the entry condition of the orientation distribution to be set at the gate, so that when kappa = 0, the RSC returns to the Folgar–Tucker model.

Phelps and Tucker developed an Anisotropic Rotary Diffusion (ARD) model that couples RSC to a new six-parameter ARD-RSC model for constant fiber orientation prediction [17]
(10)$$\begin{array}{ll} {\overset{\&}{A}}^{\mathrm{ARD}-\mathrm{RSC}} & =W\cdot A-A\cdot W+\xi \{D\cdot A+A\cdot D-2[A_{4}+(1-\kappa )(L_{4}-M_{4}:A_{4})]:D\}\\ & +\overset{\&}{\gamma }\{2[D_{r}-(1-\kappa )M_{4}:D_{r}]-2\kappa \mathrm{tr}(D_{r})A-5(D_{r}\cdot A+A\cdot D_{r})\\ & +10[A_{4}+(1-\kappa )(L_{4}-M_{4}:A_{4})]:D_{r}\}. \end{array}$$

### Different Materials Relationship and Effects

It can be seen in Figure 2a,b, these two PvT charts of PP raw material and glass fiber that the crystalline abnormality point of pure PP raw material is about 126 degrees, no matter how the internal pressure of the cavity changes, even with the large addition of 30% glass fiber to the PP raw materials. Due to the difference in pressure, the transformation of the crystalline point exhibits abnormal behavior from 126 degrees to 184 degrees because of the change in its specific volume pressure, where the difference is as large as 60 °C. In other words, it can only be determined from the PvT diagram whether the PP raw material had glass fiber added (if the manufacturer adds a second recycled PP raw material, the PvT diagram will be different).

After determining the addition of materials to the PP raw materials in the PvT diagram, starting from the curing temperature of the gate (in Figure 2a, before 126 degrees, about 115–120 degrees), under different conditions, cavity internal pressure (holding pressure) can be known. Assuming that the internal pressure of the cavity is 100 MPa (the green curve) as the ideal holding pressure, then we can infer that the internal pressure of the cavity is close to 0 MPa (the brown curve) due to the insufficient holding pressure, and the internal pressure near 200 Mpa (the purple curve) is the over holding pressure. The experimenter can find the proper holding pressure amongst these three curves, and the set value will not deviate much.

## 5. Warping Alignment Control with the Simulation Model

The same model for the fiber discharge residual experiment is used for the experimental warpage deformation, by being poured into the side runner for experiments. The plastic also uses PP without glass fiber and three different holding pressure times to find the processing parameter relationship between warpage and deformation and holding pressure time. The output information is the mold flow deformation analysis, which is then compared to the effect of different warpage deformations on the structural orientation. This analysis also shows that the holding pressure time improves the warpage deformation during the mold flow analysis.

### 5.1. Warping Interaction Effects

Warpage deformation is a common defect in plastic products; it refers to a phenomenon whereby plastic products deviate from the intended design shape due to surface distortion. Excessive warpage deformation has a serious impact on the product’s appearance quality and user performance, and even leads to a product being scrapped. In the evaluation demand requirements, it is very likely that the interaction effect warpage and deformation only occur in specific situations, and do not increase the demands under specific factors. In terms of the usability evaluation, reduced conspicuous interaction affects warpage and deformation is an aid in a product’s characteristics. Although the occurrence of an interaction effect does not mean that a single independent variable is not a single factor affecting the dependent variable, we can still discuss what the interaction reveals when it occurs.

Interaction is a research factor that must be considered. The traditional Taguchi method is aimed at continuous improvement, but the purpose is not related to interaction. Although some articles mention VSM, they are not very diligent on this issue, and only the following definitions are made in physics:

Interaction is a kind of action that occurs as two or more objects have an effect upon one another. The idea of a two-way effect is essential in the concept of interaction, as opposed to a one-way causal effect. Closely related terms are interactivity and interconnectivity, of which the latter deals with the interactions of interactions within systems: combinations of many simple interactions can lead to surprising emergent phenomena [18].

The concept of the interaction effect is explained in physics as when two or more independent variables exist in an experimental study, and the effect of one of the independent variables exhibits an alternating phenomenon on the other independent variable. The true effect of some factors is changed by another factor. The variables separable model (VSM) in Taguchi’s method is used to evaluate this interaction effect. It can be used to quantify the interaction between the values of the warping orientation of glass fibers. This method provides reference data and provides a simple way to determine the amount of impact on the molded object depending on whether or not the glass fiber was added and the holding pressure time condition [15,16,17].

This proposed comparison of warping alignment is in accordance with the interaction effect value η from the VSM method, shown as: *η*(*A*,*B*,*C*,…) = *η* + *a*(*A*) + *b*(*B*) + *c*(*C*) + …(11)

According to the VSM method, the VSM model is applicable to evaluate the interaction effect between different effect values:*η*(*Ai* + *Bj* + *Ck*…) = *η*(*ηAi* − *η*) + (*ηBj* − *η*) +(*ηCk* − *η*) + …(12)

In statistics, variables are usually classified into several categories according to the characteristics of the variables. These variables have different levels. Categorical variables are lowest-level variables and can only be used to distinguish things. The order variable is one level higher than the categorical variable. In addition to distinguishing between different things, it can also sort order and size, etc. The most important thing here is that it can do statistical operations to meet the use of big data.

### 5.2. The Results of the Simulation Analysis of the Forming Issue Are Summarized as

Warpage deformation is a common defect in plastic products; it refers to a phenomenon whereby plastic products deviate from the intended design shape due to surface distortion. Excessive warpage deformation has a serious impact on the product’s appearance quality and user performance, and even leads to a product being scrapped. In the evaluation demand requirements, it is very likely that the interaction effect warpage and deformation only occur in specific situations, and do not increase the demands under specific factors. In terms of the usability evaluation, reduced conspicuous interaction effect warpage and deformation is an aid in a product’s characteristics. Although the occurrence of an interaction effect does not mean that a single independent variable is not a single factor affecting the dependent variable, we can still discuss what the interaction reveals when it occurs.

(1)In the plastic injection molding process, the addition of glass fibers increases the structural strength, and the higher the glass fiber content, the stronger the structural strength. With regards to the orientation of the injection molding flow, when there are more glass fibers arranged in a row, the support strength is greater. If the molded product has more glass fibers arranged in the direction of the stress pressure, the structural strength will be worse.(2)Increasing the holding pressure time can reduce the residual stress of the injection-molded product. The analysis results show that residual stress is greatest at the side of the gate. Greater residual stress increases the stress of the overall structure when it is under force, and so more likely leads to structural damage.(3)Increasing the holding pressure time can reduce the warpage and deformation of the injection-molded product. The larger the warpage deformation, the greater the stress generated when the structure is under force. Thus, the deformation of the molded product is not only affected by the design accuracy, but also by the structural strength.

The total warpage simulation analysis results are shown in Figure 3. It can be seen that there is a combination of warpage direction in the model, but the main structure is not subject to force. Therefore, this configuration design is still acceptable, and the glass fiber orientation alignment in the second case is mostly the same. The residual deformation stress orientation analysis results are quite different.

From the above figure, we can determine that pure PP raw material is different from PP with additional glass fibers. Simply speaking, the warping orientation is the biggest difference. Figure 3 shows the amount of warpage deformation of pure PP raw materials, while the right figure shows the amount of warpage deformation of PP raw materials with glass fiber added. Therefore, the difference in the specific volume between liquid and solid state is one of the reasons for shrinkage after plastic processing, and the specific volume of plastic is a function of phase state, temperature, and pressure. The PvT diagram during the holding analysis shows that during the holding phase, the pressure inside the mold cavity climbs rapidly, and the shrinkage, warpage, and residual all have corresponding effects on the change in specific volume.

## 6. Case Study

The complete process parameters of traditional injection molding include around 20–30 strokes, and a larger system can even reach more than 40 strokes, but not all of these parameters affect the main focus of defect formation in products. If the forming defects are differentiated according to the display theory, the key one to three impact factors are sufficient to represent the improvement options for manufacturing in stages. In the fourth section, we implemented an injection molding process simulation, and we learned that the main improvement options relate to the glass fiber content and the holding time. Therefore, we assume that these two options are set as the main experimental influence factors. Warpage displacement data were obtained through the actual injection molding process.

There are many kinds of warpage displacement of products. It is more common to use reverse engineering technology to measure the surface point data, and then the data can be generated to obtain the warpage rate detection.

After inputting the combined parameter factors into the analysis parameter settings, the output provides the individual conditions encountered during the import process; the analysis results during the process are shown in Table 1. It can be seen that the first group’s warpage rate is the largest. From Table 1, we can determine that the products with and without glass fiber basically display opposite warpage trends. At the same time, it is found that the warpage trends of groups 1, 4, and 7 are the same, and that the warping trends of groups 2, 3, 5, 6, 8, and 9 are completely different. It can be seen that in the parameter range that we set, the glass fiber content is the most important parameter factor.

The trend of holding pressure time is basically consistent with the amount of glass fiber content, but we cannot determine the proportion of its consistency, that is, the interaction effect value cannot be digitized. In the fourth section, we can determine that the addition of glass fiber will directly affect the displacement orientation of the warped area. Therefore, the warpage displacement data is only an experimental phase result; in the experiment the warpage data is mixed, we cannot determine know how product deformation is affected.

The degree of slope in the three lines of the first group of Figure 4 is larger, which means that the product without glass fiber is unstable relative to the product with glass fiber, and the interaction effect value generated was high regardless of the holding pressure. In addition, Figure 5 has a uniform three-line trend, so the force trend is the same, and the warping ratio is similar. This graph can also be used to determine whether the raw materials in the process are secondary recycled materials.

Figure 6 shows that the glass fiber content of all the experimental objects is zero, and the relative relationship between the deformation amount and the interaction effect value is non-parallel but not intertwined. In addition, in Figure 7 and Figure 8, the glass fiber content is 20 and 30, respectively, and the relative relationship between the deformation and the interaction effect is non-parallel and intertwined, especially for GF = 30 when the degree of slope error is almost 60–90 degrees.

This also means that the glass fiber content and the holding pressure parameters have more interaction effects. Conversely, products without glass fiber content can only be adjusted through holding pressure; regarding the product on the Figure 9a, when the geometric design of the connection side is too small or difficult to flowing, the actual situation is prone to warp, and modifying the hold pressure parameter to be modified and is necessary.

Hence, when the glass fiber content is higher, the uniform effect of holding pressure theoretically becomes more significant. In addition, the product on the Figure 9b has a certain amount of glass fiber that can improved geometric shape and reduced warpage, so it can be known from practice that this method is achievable.

## 7. Conclusions

The analysis of the amount of glass fiber added and the holding pressure time shows the impact of molding warpage and deformation problems of structural parts based on the information integration of the reverse warpage experimental factors for glass fiber injection molding. We know that the addition of glass fiber can effectively increase the structural strength of the part; if the glass fiber content is higher, the flow during filling can be controlled by the design of the gate, which can increase the structural strength of the part. Increasing the holding pressure time can reduce the warpage deformation and residual stress of the injection-molded product, and thereby increase the structural strength of the molded product. At the same time, the effect of warping orientation is directly related to the addition of glass fiber.

The interaction analysis and integration process established in this paper can only control the molding parameters to achieve optimized improvement without changing the geometric shape of the molded product, which greatly reduces the cost of repairing or reopening the mold. The integration process proposed in this paper can achieve structural warpage improvement in an innovative method; in the past, the finite element analysis software was used to evaluate structural strength, but the only area that this older method can provide improvements to designers is by changing materials or by replacing geometric shape designs. Integrated analysis can adjust the configuration parameters in injection molding to achieve increased stability, which improves the reliability and quality of injection-molded products.

## Figures and Tables

**Figure 1 materials-13-01151-f001:**
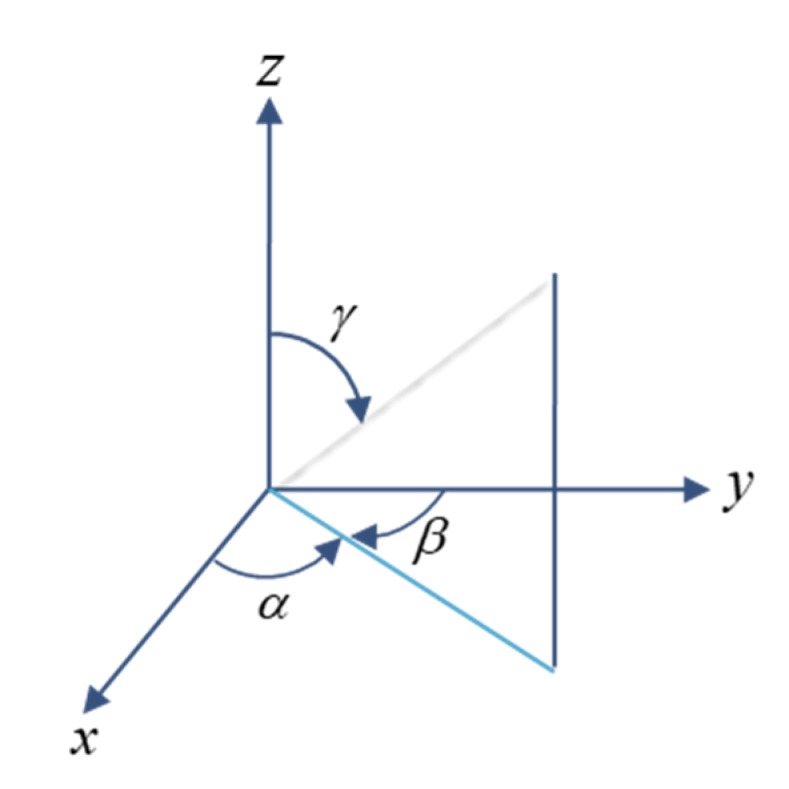
Definition of Fiber Orientation.

**Figure 2 materials-13-01151-f002:**
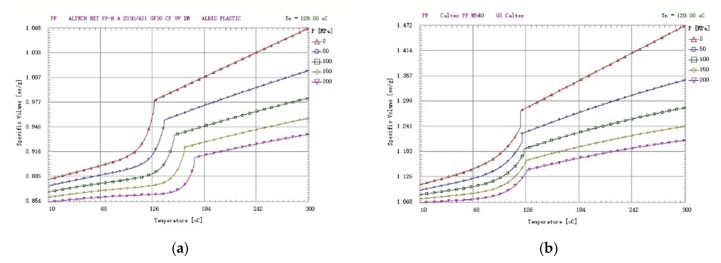
The PvT chart with PP material and different Glass Fiber.

**Figure 3 materials-13-01151-f003:**
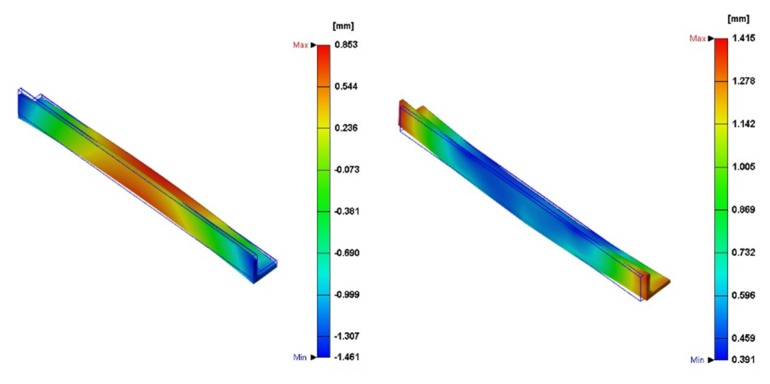
Warpage orientation simulation analysis results with different glass fiber [17].

**Figure 4 materials-13-01151-f004:**
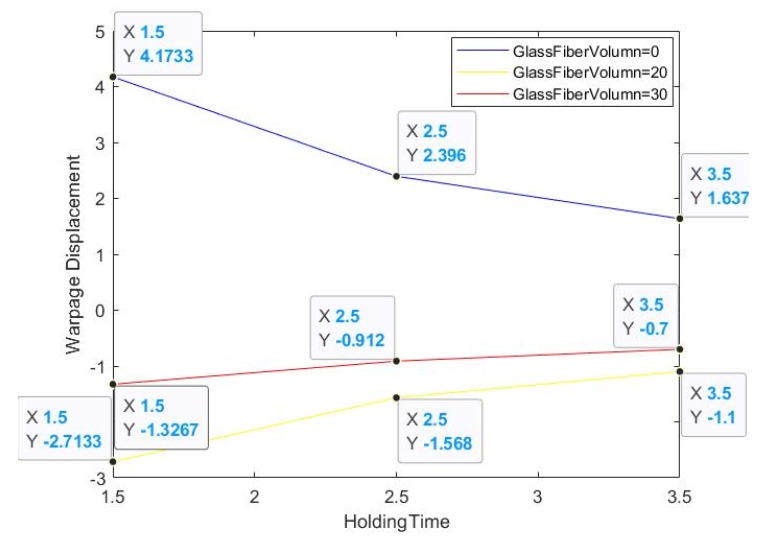
Warpage displacement with glass fiber and holding.

**Figure 5 materials-13-01151-f005:**
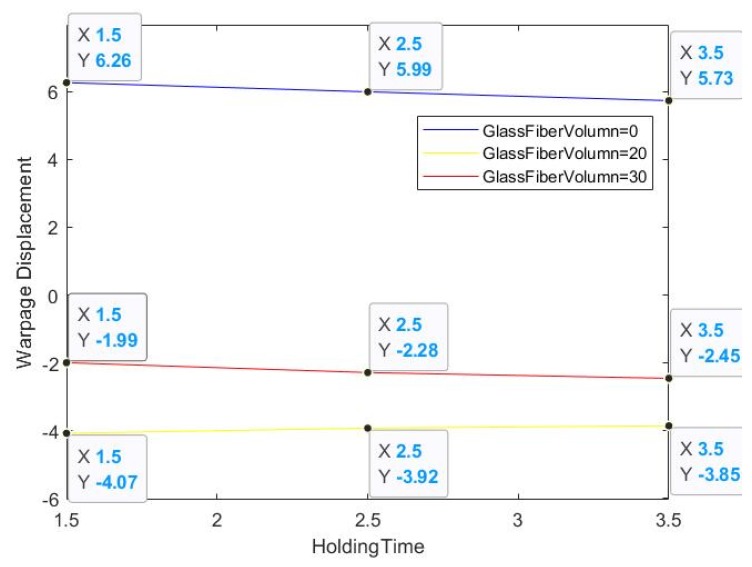
Warpage displacement with glass fiber and holding.

**Figure 6 materials-13-01151-f006:**
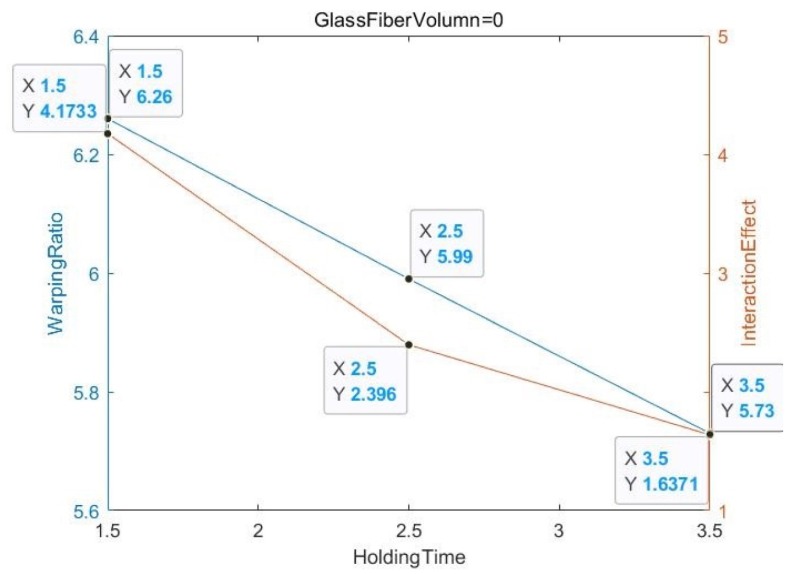
GF Volume = 0.

**Figure 7 materials-13-01151-f007:**
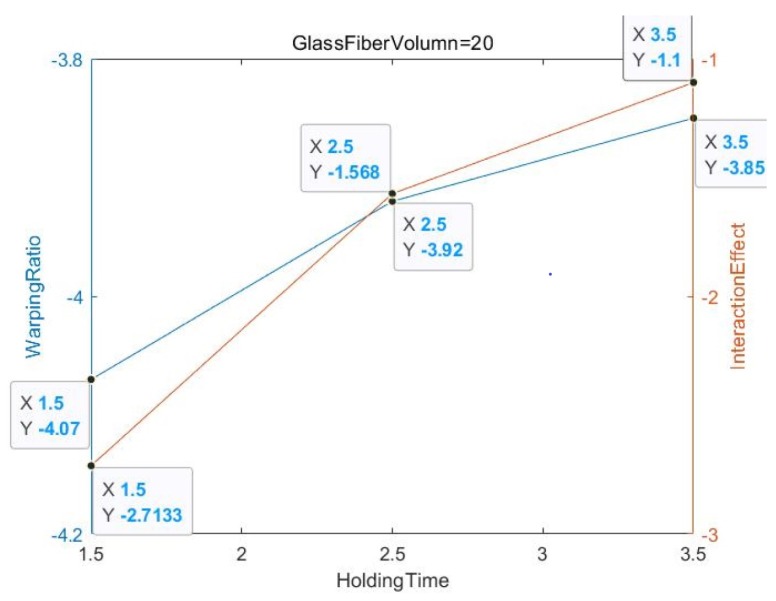
GF Volume = 20.

**Figure 8 materials-13-01151-f008:**
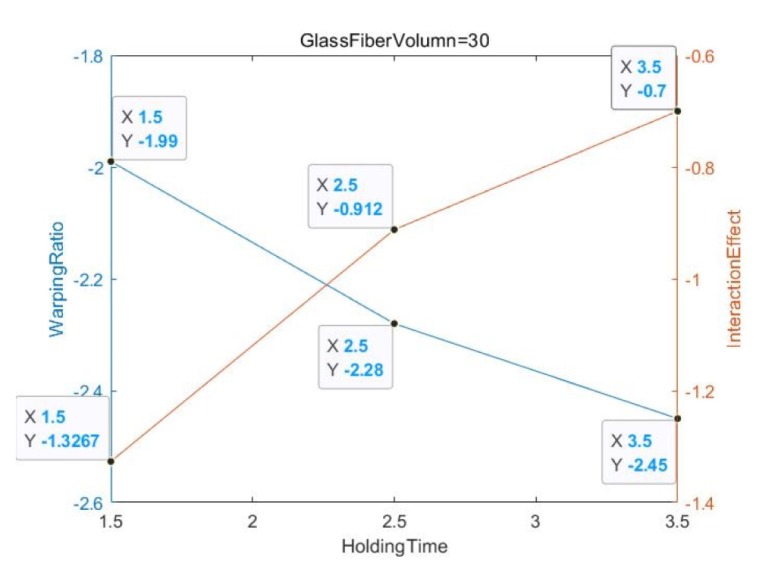
GF Volume = 30.

**Figure 9 materials-13-01151-f009:**
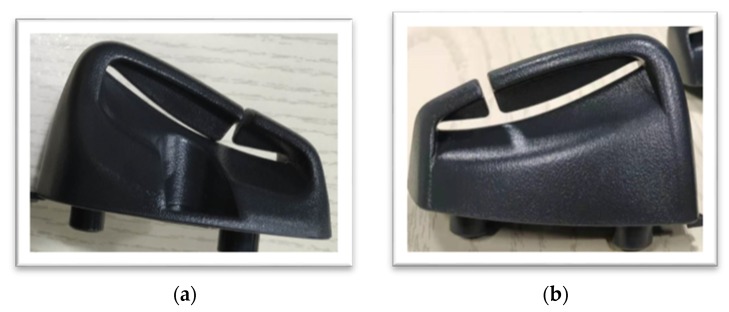
Improving Experiment with different GF and Holding.

**Table 1 materials-13-01151-t001:** Relationship of Interaction effect and Warping displacement.

No.	Holding Time (s)	Glass Fiber Volume (%)	Warping Displacement (%)	Interaction Effect
1	1.5	0	6.26	4.17
2	2.5	20	−3.92	−1.57
3	3.5	30	−2.45	−0.70
4	2.5	0	5.99	2.40
5	3.5	20	−3.85	−1.10
6	1.5	30	−1.99	−1.33
7	3.5	0	5.73	1.64
8	1.5	20	−4.07	−2.71
9	2.5	30	−2.28	−0.91

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
