# Peer review of "Optimizing Glass Fiber Molding Process Design by Reverse Warping"

_materials, 2020, doi:10.3390/ma13051151_

Round 1
Reviewer 1 Report
In Abstract, the main contents should be revised. Author should give an adequate description of the main purpose and scope of the study. In addition, it should contain the description of the methodology chosen for research purposes. Finally, it should main findings and conclusions. The author should revise the abstract continuing main points mentioned above in the precise way. In the introduction section, the author gives an adequate overview about the state of knowledge describing the necessary background and information. However, in the current state the motivation of the study is missing. The author should point out the general motivation of this study. In addition, the main point that this article aim to explain should be mentioned more sententiously. In the simulation of warping Alignment Control with the Simulation Model, author should present detailed simulation conditions such as detailed numerical methods, initial and boundary condition, method for the mesh creation and grid information should be presented. This part should be reformulated. The important stage for the numerical simulation is omitted in the paper. Dependency results confirming the solution should be shown. In warping Alignment Control with the Simulation Model section, some sentences should be revised due to containing ambiguous meaning. For example, the meaning of “with regards to the orientation of the injection molding flow, when there are more glass fibers arranged in a row, the support strength is greater. If the molded product has more glass fibers arranged in the direction of the stress pressure, the structural strength will be worse.” is not clear.And the exact manning of the simulation is hard to understand without the legends, and unclear meaning of caption of figures especially for figure 1. In Warping interaction effects section, interaction effect warpage and deformation should be revised with more detailed explanation containing specified results. With these descriptions, the meaning of the usability evaluation and reduced conspicuous interaction effect would be understood. The meaning in the sentence of “Although the occurrence of an interaction effect does not mean that a single independent variable is not a single factor affecting the dependent variable, we can still discuss what the interaction reveals when it occurs.“ in Warping interaction effects section should be revised in more concise way. In Case Study section, the meaning of stroke and 80-20 theory is not clear. It should be added some detailed descriptions for the meanings. The main results of this article present “with less GF volume fraction and less holding time, large the warping displacement”. However, it is natural that such a result will occur. Besides the results, it should be add more reasons and descriptions for the reasons. In Case Study section, the meaning of results is explained more detail: the problem can be improved if additional work is performed to explain interaction effects.
Author Response
Responds:
Appreciate for valuable advise, please see new manuscript which have revised regarding abstract and research purposes description. Additional work also includes warping of the effects of exhalation interactions. besides, the English modification by British native speaker.

Reviewer 2 Report
Several major revisions are needed for this paper to be published.
There is a need to clarify the contribution of this paper. Is there any improvement about model used in this paper compare to previous study? Hasn't there been any previous study of GFRP warpage using the Taguchi (VSM)? More detailed explanation about VSM is needed
- Page6 (VSM) A detailed description of each of the variables and what they mean in this study.
It is necessary to specify detailed information about the software used for simulation. (version, company) Figure1. Detailed information on the model applied to this simulation and the criteria for warpratio measurement is required Page6 There is a typo(Oo Ss) in Figure 1. In table1, detailed description and analysis of the formula about how the interaction effect value was calculated is required.
Author Response
Response:
Thanks for your valuable feedback,
Concerning the contribution of this paper, there is not much research on Taguchi (VSM) for GFRP warping in the past, especially the interaction between holding pressure and warping direction of glass fiber raw materials.
In addition, the English modification by British native speaker.
Reviewer 3 Report
The manuscript presents a parameter study in the field of injection moulding with glass fibre filled plastics. The main objective of the study is to scrutinise the influence of material and process parameters, namely the fibre volume fraction and the holding pressure time, on the warping of the part using the Taguchi method. The conclusion gives some advice on how to manipulate the parameters in order to achieve less warpage. Furthermore, interactions between the two scrutinised parameters are evaluated.
The idea of improving the product quality by using integrative simulation and optimisation provides an important contribution to the development and design of fibre reinforced injection moulded parts. The authors should be encouraged to keep on working on this topic. While the title and the abstract announce an optimisation method based on reverse warping, the article lacks in presenting the method. Instead, results of a parameter study are presented. Please improve the paper by adding and describing the method (more detailed), e.g. by using flow charts, as well as experimental setups. Furthermore highlight the novelty of the presented work compared to the state of the art more clearly.
The second main criticism is the structure, language and format of the article. The sections and chapters are incoherent. Please try to improve the structure in a more logical way for example by sticking to the recommendations made in the template.
Beside these main issues, further annotations regarding the content and the formal aspects are made.
Specific annotations regarding the content:
The second paragraph of the Introduction describes parts of the used method. Please move this to the method section. Subsectioning of the literature review is recommended. Thematic sections might be injection moulding/warpage, reverse warpage, optimization and Taguchi method Line 85-89: relevance? Line 95-102: relevance? Line 139-141: Is it an official definition? Reference? Chapter 3: Addition of references in first paragraph are suggested First to second paragraph harsh break. Short introduction to the used method and the in the following described fibre orientation control and its integration in the method should be supplemented. Line 178-183: Please add some references Line 184: Which measures are the different symbols in the formula? Chapter 4 starts with a summary of simulation analysis results but no simulation analysis was described or presented. Where do the results come from? Please describe the simulative models and experiments used for analysis. Figure 1 is not described properly: The flow wavefronts can not be seen. A scale, including the shown measure, is missing. The caption lacks of the information which image shows which glass fibre content. VSM from Taguchi method is described quite short. Please enhance the description. Line 271-272: Can this be seen in the results? Where is the connection to the papers topic? Line 283: What is meant with degree of slope error is almost 60-90°? Could the authors comment on this?
Formal annotations:
Properly check whether your manuscript is consistent with format rules of the template.
Improve quality of Figures (especially Figures 2 - 6 are of bad quality) Figure captions are very short. Please describe the content. When having more pictures in one Figure divide into subfigures a), b), … Add caption to Figure of the fibre orientation A proper Spell/Language check is strongly recommended Number the Equations Improve (picture) quality of Equation (4) and (5) Check for the explanation of abbreviations when using the abbreviation for the first time Add DOIs Figure 7 is not referred to
Summarising all deficiencies, the contribution should be rejected. Nonetheless the reviewer wants to encourage the authors to resubmit the manuscript after fundamental revisions.
Author Response
Response:
Thank you for valuable feedback,
@Line 85-89 and Line 95-102 already revised, and give Line 139-141 reference definition.
@New manuscript which has revised regarding abstract and research purposes description. Additional work also includes warping of the effects of exhalation interactions.
@Revised flow wavefronts to total warpage and warpage direction.
@Line 271-272 and Line 283 already more additional description.
@Format rules of the template has revised, thank you.
@Improve quality of Figures (especially Figures 2-6 are of bad quality), has revised, thank you.
In addition, the English modification by British native speaker.
Round 2
Reviewer 1 Report
The papaer has been reworked as it was recommended. However, for the Figure 1, it needs to be revised to get visibility.
Author Response
Responds:
Appreciate for valuable advise, already change Figure 1 as commended, thank you.
Reviewer 2 Report
I think it would be good to change the axis name of Figure 1 to English. There is a typo in Figure 2. You have two Figure 3. There is a typo in Figure 3. In page 7(VSM), a detailed explanation of the variables is needed. In page 9, the criteria for warping ratio measurement is still insufficient.
Author Response
Responds:
Appreciate for valuable advice,
@already change Figure 1 as commended, thank you.
@ already revised the typo of Figure 2, thank you.
@ already revised the typo of Figure 3, thank you.
@ already increased explanation of the variables as recommendation, thank you.
@ already increased more about the warping ratio measurement as recommendation, thank you.
Round 3
Reviewer 1 Report
The author reworked the whole article as it was recommended. All of the comments were considered or discussed.
Author Response
Responds:
Appreciate for valuable advise.
Reviewer 2 Report
The paper could be published after some minor revisions.
- The y-axis name(warpingratio) in Figure 3 was not modified.
- You have two Figure 7.
- Other points seem to be properly corrected.
- If there's anything else, it would be better to classify the contents (line 223-238) as sub-headings.
Author Response
Responds:
Appreciate for valuable advice,
@ already modified Figure 3 as commended, thank you.
@ already modified Figure 7 and 8, thank you.
@ already created sub-headings for the contents (line 223-238) as commended, thank you.